# Copper Oxide Nanoparticles Induce Pulmonary Inflammation and Exacerbate Asthma via the TXNIP Signaling Pathway

**DOI:** 10.3390/ijms252111436

**Published:** 2024-10-24

**Authors:** Woong-Il Kim, So-Won Pak, Se-Jin Lee, Sin-Hyang Park, Je-Oh Lim, In-Sik Shin, Jong-Choon Kim, Sung-Hwan Kim

**Affiliations:** 1College of Veterinary Medicine and BK21 FOUR Program, Chonnam National University, Gwangju 61186, Republic of Korea; dvmwoong@gmail.com (W.-I.K.); dvmpsw@gmail.com (S.-W.P.); xhdhksdl123@naver.com (S.-J.L.); shinhyang23@gmail.com (S.-H.P.); dvmmk79@gmail.com (I.-S.S.); 2Herbal Medicine Resources Research Center, Korea Institute of Oriental Medicine, Naju 58245, Republic of Korea; dvmljo@kiom.re.kr; 3Jeonbuk Department of Inhalation Research, Korea Institute of Toxicology, Jeongup 56212, Republic of Korea

**Keywords:** copper oxide nanoparticles, pulmonary toxicity, asthma, thioredoxin-interacting protein

## Abstract

Copper oxide nanoparticles (CuO NPs) have seen increasing use across various industries, raising significant concerns about their potential toxicity and the exacerbation of pre-existing conditions like asthma. Asthma, a chronic inflammatory condition of the airways, can be triggered or worsened by environmental factors such as allergens, air pollutants, and chemicals, including nanoparticles. This study aimed to investigate the pulmonary toxicity induced by CuO NPs and their impact on asthma, with a particular focus on the role of thioredoxin-interacting protein (TXNIP). Using an ovalbumin (OVA)-induced asthma model, we found that CuO NP exposure led to significant increases in inflammatory cell infiltration, cytokine production, airway hyperresponsiveness, OVA-specific immunoglobulin (Ig)E levels, and mucus production. These pathological changes were closely associated with the upregulation of TXNIP-related signaling pathways, including phosphorylated apoptosis signal-regulating kinase (p-ASK)1, the Bax/Bcl-2 ratio, and cleaved caspase-3 activation. Complementary in vitro experiments using NCI-H292 respiratory epithelial cells showed that CuO NP treatment enhanced TXNIP signaling and increased mRNA expression and the production of inflammatory cytokines. Notably, TXNIP knockdown significantly attenuated these CuO NP-induced effects. In conclusion, our findings suggest that CuO NP exposure not only induces pulmonary toxicity but also exacerbates asthma, primarily through the activation of the TXNIP signaling pathway.

## 1. Introduction

Copper oxide nanoparticles (CuO NPs) have outstanding physicochemical, photoconductive, and antimicrobial properties, leading to their widespread application in a variety of industrial fields [1,2,3]. They are utilized extensively in medicine for protein biomarker detection, tumor treatment, and antimicrobial activities, as well as in agriculture for fertilizer and wood preservation, and in engineering for energy storage and microelectronics [4,5,6]. As the applications of CuO NPs gradually expand, concerns regarding environmental pollution near emission sources and adverse effects on human health are increasing. This has led to emerging studies on the potential toxicity of CuO NPs through various toxicological evaluations [7,8]. CuO NPs are known to exhibit stronger potential toxicity when inhaled compared with other metal nanoparticles [9,10]. Studies have reported that inhaled CuO NPs are effectively absorbed, resulting in elevated levels of reactive oxygen species (ROS) in the body, and ultimately inducing oxidative stress and cell death [7,11]. Additionally, these nanoparticles can induce genotoxicity and cytotoxicity, as well as directly activate pro-inflammatory proteins and transcription factors, and cause a wide range of inflammatory responses and cell death [12,13,14]. Furthermore, as interest in the toxicity of CuO NPs grows, recent studies have investigated how exposure to CuO NPs exacerbates underlying pulmonary diseases [15,16]. Nevertheless, research on the toxic mechanisms underlying the potential toxicity induced by exposure to CuO NPs and their effects on underlying pulmonary diseases remains very limited.

Asthma is one of the most prevalent respiratory diseases globally, affecting approximately 330 million people, with its prevalence increasing by 50% every decade [17]. Asthma is induced by repeated allergic and airway hypersensitivity reactions caused by various irritants, such as allergens and air pollutants, leading to chronic inflammation [18]. This results in airway remodeling, thickening of the smooth muscles, and increased contractility, which induces clinical symptoms such as coughing, shortness of breath, and chest pain [19,20]. With the recent gradual increase in air pollution, exposure to various particles, including nanoparticles, has accelerated the development and exacerbation of asthma. Consequently, research has demonstrated a link between the development of asthma and exposure to external substances, particularly nanoparticles [21,22,23,24]. However, the toxic mechanisms through which nanomaterial exposure affects asthma are not yet fully understood.

Thioredoxin-interacting protein (TXNIP) is involved in the pathophysiological alterations of several disorders. It inhibits thioredoxin (TRX), a crucial regulatory protein in all organisms, and triggers inflammatory responses, immune responses, and cell death [25,26]. Notably, TXNIP is known to induce apoptosis in response to oxidative stress. Under stable conditions, TRX binds to the end of apoptosis signal-regulating kinase1 (ASK1) and inhibits ASK1-dependent apoptosis. Conversely, under oxidative stress, TXNIP can bind to TRX in the cytoplasm or mitochondria, leading to the release of ASK1. This release triggers ASK1-dependent signaling, leading to the cleavage and activation of caspase-3 and subsequent apoptosis [27,28]. Apoptosis is a programmed cell death process that depends on the activation of the caspase cascade, and cleaved caspase-3 has been recognized as a key mediator of apoptosis, as it is essential for key processes involved in cell degradation and programmed cell death [29,30]. Furthermore, in the respiratory system, TXNIP is expressed in the lungs of asthmatic mice. Notably, exposure to nanoparticles, such as silica dioxide and titanium dioxide nanoparticles, significantly increases TXNIP expression in asthmatic mice, leading to enhanced pulmonary toxicity [23,31]. However, research examining the correlation between CuO NPs, asthma, and TXNIP is lacking, and there is a dearth of studies elucidating the role of TXNIP in the development and exacerbation of asthma caused by CuO NP exposure.

To address this gap in the literature, this study aimed to examine the respiratory toxicity of CuO NPs using in vivo and in vitro models. Building on these results, we evaluated the impact of CuO NPs on the development and exacerbation of asthma, with a particular emphasis on a novel pathway, the TXNIP/ASK1 pathway, which has not been explored in previous studies on respiratory toxicity induced by exposure to CuO NPs, providing a new perspective on targeted signaling strategies.

## 2. Results

### 2.1. Physicochemical Characteristics of CuO NPs

CuO NPs exhibited a typically spherical form, as analyzed using SEM and TEM. The primary size of the CuO NPs measured using SEM was 41.66 ± 13.28 nm (Figure 1). Energy-dispersive X-ray spectroscopy identified the material as CuO, with an atomic ratio close to 1, consisting of Cu 43.81% and O 56.19%. Using ELSZeno, the hydrodynamic size of CuO in phosphate-buffered saline (PBS) and its zeta potential were determined to be 594.2 ± 109.7 nm and −20.54 mV, respectively.

### 2.2. Effects of CuO NPs on the Inflammatory Factors of Bronchoalveolar Lavage Fluid (BALF) and Histopathological Changes in Mice

In the pulmonary toxicity experiment involving CuO NPs, the lowest dosage group did not exhibit a significant increase in the number of inflammatory cells or the production of inflammatory cytokines in BALF compared with the normal control (NC) group. However, exposure to CuO NPs at doses above 0.4 mg/kg significantly elevated the number of inflammatory cells, including neutrophils, macrophages, and lymphocytes, as well as the levels of inflammatory cytokines such as interleukin (IL)-1β, IL-6, and tumor necrosis factor (TNF)-α compared with the NC group (Figure 2A–G). Mice treated with over 0.2 mg/kg of CuO NPs displayed elevated mucus production and an accumulation of inflammatory cells surrounding the bronchi and alveoli compared with the NC group (Figure 2H–J).

### 2.3. Effects of CuO NPs on TXNIP and Apoptosis-Related Signal Expression in Mice

Exposure to CuO NPs at doses above 0.2 mg/kg remarkably increased TXNIP expression compared with the NC group (Figure 3A,B). Compared with the NC group, CuO NP exposure significantly elevated apoptosis-related signals, including TXNIP, phosphorylated ASK1 (p-ASK1), Bax, and cleaved caspase-3, while considerably decreasing Bcl-2. This occurred in a dose-dependent manner (Figure 3C–I).

### 2.4. Effects of CuO NPs on Airway Hyperresponsiveness (AHR), Inflammatory Factors, and Immunoglobulin (Ig)E Levels in Asthmatic Mice

Compared with mice in the NC group, the mean AHR levels in the ovalbumin (OVA) group were significantly higher at all methacholine concentrations. Additionally, exposure to CuO NPs at doses exceeding 0.5 mg/kg significantly raised the average AHR level in asthmatic mice compared with the OVA group (Figure 4A). The total number of inflammatory cells in the lungs of the OVA group was considerably higher than in the NC group, and this increase was even more pronounced in OVA-induced asthma mice exposed to more than 0.5 mg/kg of CuO NPs (Figure 4B). The elevation was particularly notable in macrophages, neutrophils, and lymphocytes. In the CuO NP 1.0 group, eosinophil counts also increased significantly (Figure 4C–F). Among the pro-inflammatory cytokines, the levels of IL-4, IL-5, and IL-13 were remarkably elevated in the OVA group compared with the NC group and increased further in the OVA + CuO NP 1.0 group compared with the OVA group (Figure 4G–I). Additionally, the levels of IL-1β, IL-6, and TNF-α did not significantly differ between the NC and OVA groups but were remarkably higher in the OVA + CuO NP 1.0 group compared with the OVA group (Figure 4J–L). Furthermore, the CuO NP-exposed groups exhibited higher levels of total IgE and OVA-specific IgE than the OVA group (Figure 4M,N).

### 2.5. Effects of CuO NPs on Histopathological Changes in Asthmatic Mice

To examine the accumulation of inflammatory cells and mucus secretion, a histopathological analysis using H&E and PAS staining was conducted (Figure 5A). The OVA group exhibited a notable increase in the accumulation of inflammatory cells compared with the NC group. This increase was more pronounced in all OVA + CuO NP-treated groups, with the extent increasing with increasing CuO NP concentrations (Figure 5B). Similarly, mucus secretion was significantly higher in the OVA group compared with the NC group. Furthermore, OVA-induced asthmatic mice exposed to more than 0.5 mg/kg of CuO NPs showed higher mucus secretions compared with mice in the OVA group (Figure 5C).

### 2.6. Effects of CuO NPs on TXNIP and Apoptosis-Related Signals in Asthmatic Mice

The expression of TXNIP and cleaved caspase-3 was demonstrated through immunohistochemistry (IHC). Mice in the OVA group exhibited higher levels of TXNIP and cleaved caspase-3 expression compared with the NC group. When OVA-induced asthma mice were exposed to more than 0.5 mg/kg of CuO NPs, these levels were significantly higher than those in the OVA group (Figure 6A–C). Additionally, a terminal deoxynucleotidyl transferase dUTP nick-end labeling (TUNEL) assay was conducted to examine alterations in lung parenchymal apoptosis. A noticeable increase in apoptotic cells was observed in the OVA group compared with the NC group, and this increase was more pronounced when OVA-induced asthmatic mice were exposed to more than 0.5 mg/kg of CuO NPs (Figure 6A,D). Furthermore, TXNIP and apoptosis-related signals were analyzed using the Western blot method. Mice in the OVA group exhibited higher levels of TXNIP, p-ASK1, Bax, and cleaved caspase-3 expression compared with the NC group, and OVA-induced asthma mice exposed to CuO NPs indicated higher levels of expression than mice in the OVA group. Conversely, the expression level of Bcl-2 was highest in the NC group and decreased as the concentration of CuO NPs increased (Figure 6E–K).

### 2.7. Effects of CuO NPs on Cell Viability, mRNA Expression Levels, and Amounts of Inflammatory Cytokines in NCI-H292 Cells

Based on the cell viability assay, the maximum concentration of CuO NPs for the in vitro experiments was set at 2.0 µg/mL, as higher concentrations resulted in significant cytotoxicity (Figure 7A). In cells treated with CuO NPs, the mRNA expression levels of TNF-α, IL-1β, and IL-6 were significantly elevated in a dose-dependent manner compared with untreated cells (Figure 7B–D). Upon assessment in the cell suspension, the levels of IL-6 and IL-8 were significantly elevated in cells treated with more than 0.5 µg/mL of CuO NPs compared with untreated cells (Figure 7E,F).

### 2.8. Effects of CuO NPs on TXNIP and Apoptosis-Related Signals Expression in NCI-H292 Cells

CuO NP-treated cells exhibited remarkably increased expression of TXNIP, p-ASK1, Bax, and cleaved caspase-3 compared with untreated cells, while considerably decreasing Bcl-2 (Figure 8A–G). Additionally, double-immunofluorescence staining revealed a marked accumulation of TXNIP and p-ASK1 in CuO NP-treated cells (2.0 µg/mL) compared with untreated cells. This accumulation was comparable with that observed in hydrogen peroxide-treated cells (100 µM), which serve as a known inducer of apoptosis (Figure 8H).

### 2.9. Effects of TXNIP siRNA on TXNIP and Apoptosis-Related Signals in CuO NP-Treated NCI-H292 Cells

To investigate whether TXNIP influences the regulation of apoptotic signaling pathways in CuO NP-treated NCI-H292 cells, we conducted a functional analysis using TXNIP gene knockdown via siRNA. TXNIP siRNA treatment significantly reduced the elevated expression levels of p-ASK1, Bax, and cleaved caspase-3 induced by CuO NP exposure compared with the scrambled siRNA treatment, while the expression level of Bcl-2 was correspondingly increased (Figure 9).

## 3. Discussion

The increasing application and manufacturing of CuO NPs are significantly impacting both humans and the environment, thereby underscoring the need for research into their potential toxicity [2,7]. As the recognition of nanoparticle toxicity, including CuO NPs, has increased, studies have not only focused on their inherent toxicity but also on their effects on underlying disorders [15,23,31]. However, the toxicity of CuO NPs and their mechanisms of action in underlying pulmonary diseases remain unclear. In this study, the potential toxicity of CuO NPs and their effect on asthma were evaluated. Additionally, we analyzed the toxicological mechanism of CuO NPs, focusing on TXNIP signaling pathway. Our toxicity study revealed that exposure to CuO NPs induced neutrophilic inflammation of the respiratory tract, accompanied by elevated inflammatory cytokine levels and alterations in the histological structure, which were accompanied by the activation of the TXNIP signaling pathway and its downstream effects. These findings align with the results of our in vitro experiment. Furthermore, exposure to CuO NPs in asthmatic mice resulted in significant increases in allergic responses, including inflammatory cell counts, AHR, inflammatory cytokines, and IgE levels, compared with those in asthmatic mice.

Exposure to CuO NPs induced neutrophilic pulmonary inflammation and exacerbated allergic responses in asthmatic conditions. These results are in agreement with previous studies [15,32]. CuO NP exposure led to the production of inflammatory cytokines and ROS, which accelerated the infiltration of inflammatory cells into damaged lesions [33]. Recruited inflammatory cells, particularly neutrophils, contain amounts of inflammatory mediators in their vesicles, intensifying inflammatory responses, leading to disruption in the normal alveolar architecture, and reducing respiratory efficiency [15]. Particularly, the accumulation of neutrophils and eosinophils increased AHR and mucus production, accompanied by elevated levels of inflammatory cytokines such as IL-4, IL-5, IL-6, IL-13, IL-1β, and TNF-α. These cytokines promoted the differentiation and maturation of inflammatory cells, ultimately restricting normal airway flow in asthmatic conditions [32,34,35,36]. In this study, exposure to CuO NPs in asthmatic mice markedly increased AHR and mucus production, accompanied by the enhanced infiltration of inflammatory cells into the lung tissue, including neutrophils and eosinophils. These findings indicate that CuO NP exposure induces pulmonary inflammation and exacerbates allergic asthma by promoting inflammatory cell infiltration.

With these pathophysiological changes, exposure to CuO NPs increased TXNIP levels, leading to increased ASK1 phosphorylation, Bax expression, and cleaved caspase-3 activation, thereby increasing apoptotic changes in the lung tissues. Similar observations were also noted in the in vitro experiment, where CuO NP treatment resulted in a marked increase in the TXNIP signaling pathway and elevated levels of inflammatory cytokines. TXNIP functions as an endogenous inhibitor of TRX, a critical cellular antioxidant, thereby promoting cellular apoptosis [37]. Additionally, TXNIP is implicated in various biological alterations. It induces inflammatory responses by activating the NLRP3 inflammasome and promoting the maturation of IL-1β, while also contributing to apoptotic alteration through caspase activation [28,38,39]. Exposure to CuO NPs induces substantial production of ROS, which in turn triggers the activation of the endoplasmic reticulum stress pathway and the apoptosis pathway. In recent studies, TXNIP has been found to be upregulated as a consequence of oxidative stress, resulting in increased cellular apoptosis through the elevation of the Bax/Bcl-2 ratio and the expression of cleaved caspase-3 [28,39]. Consequently, increased oxidative stress from external stimuli induces TXNIP overexpression, inhibiting TRX and triggering NLRP3 inflammasome formation, thereby increasing caspase activation and IL-1β production. Furthermore, IL-1β activation significantly increases inflammatory cell infiltration through the activation of various immune cells, while caspase activation induces apoptotic alteration in cells within damaged lesions [40,41]. These effects have been consistently observed in numerous studies on nanoparticle toxicity [23,31]. Conversely, our in vitro experiments demonstrated that cellular toxicity induced by CuO NP exposure could be mitigated by the deletion of the TXNIP gene. These findings underscore the close association between upregulated TXNIP pathways and pulmonary toxicity, as well as the exacerbation of asthma induced by CuO NP exposure.

## 4. Materials and Methods

### 4.1. Preparation of Copper Oxide Nanoparticles

CuO NPs were obtained from Sigma-Aldrich (St Louis, MO, USA), with a specified size of less than 50 nm according to the manufacturer’s description. Prior to instillation, CuO NPs were mixed with PBS and subjected to ultrasonic processing for several minutes. The CuO NP (0.5 mg/kg) dosage was computed based on the body weight of each mouse immediately prior to administration.

Transmission electron microscopy (TEM; JEM-2100F, JEOL, Tokyo, Japan) and scanning electron microscopy (SEM; Zeiss Gemini500, Carl Zeiss Meditec AG, Jena, Germany) were used to assess the morphology and primary size of the CuO NPs at accelerating voltages of 150 kV and 15 kV, respectively. The purity of the CuO NPs was determined using energy-dispersive X-ray spectroscopy (Zeiss Gemini500 SEM equipped with X-maxN 150 mm^2^ silicon drift detector; Oxford Instruments, Abingdon, UK). The hydrodynamic size and zeta potential of the CuO NPs were measured using a particle size and zeta potential analyzer (ELSZeno, Otsuka Electronics, Tokyo, Japan).

### 4.2. Animal and Experimental Design

Specific pathogen-free BALB/C mice (female, 6 weeks old) were obtained from Samtako Co. (Osan, Republic of Korea). Upon arrival, the mice were immediately weighed and then housed three per cage to acclimate to the environment for one week prior to the experiment. The animals were maintained under controlled conditions with a relative humidity of 50 ± 5%, a 12-h light/dark cycle, 13–18 air changes per hour, and a temperature of 23 ± 2 °C. The animals were provided with a standard rodent diet and water ad libitum. All the procedures were conducted in accordance with the NIH Guide for the Care and Use of Laboratory Animals as well as following the 3R principles [42] and were approved by the Institutional Animal Care and Use Committee (IACUC) of Chonnam National University (CNU IACUC-YB-2021-72).

To evaluate the pulmonary toxicity of CuO NPs, the mice were divided into five groups (*n* = 5): one NC group and four CuO NP-treated groups (doses of 0.1, 0.2, 0.4, and 1.0 mg/kg). On days 1, 3, and 5, the CuO NP-treated groups received CuO NPs (doses of 0.1, 0.2, 0.4, and 1.0 mg/kg in 50 µL of PBS) via intranasal instillation under slight respiratory anesthesia using isoflurane (Isotory^®^, Troikaa Pharmaceuticals Ltd., Gujarat, India). The NC group received 50 µL of PBS via the same method under identical conditions. Prior to the intranasal instillation, the CuO NPs were combined with PBS and sonicated in an ultrasonicator (SD-251H, Sungdong Ultrasonic Co. Ltd., Seoul, Republic of Korea) for 3 min (150 W, 40 KHz). Mice were necropsied 48 h after the last CuO NP dose for subsequent analyses (Appendix A).

To investigate the effects of CuO NP exposure on the development of asthma, the mice were divided into five groups (*n* = 6): NC group, OVA (OVA Challenge) group, and three OVA + CuO NP groups (CuO NPs at doses of 0.25, 0.5, and 1.0 mg/kg). To induce asthma, the mice were sensitized using an intraperitoneal injection of OVA (20 μg, Sigma-Aldrich) and aluminum hydroxide (2 mg, Sigma-Aldrich) in 200 μL of PBS. On days 21, 23, and 25, the mice were subjected to airway challenges with OVA (1% *w*/*v*) for 1 h using an ultrasonic nebulizer (NE-U12, Omron, Kyoto, Japan). In the OVA + CuO NP groups, CuO NPs were administered via intranasal instillation on days 20, 22, and 24. On day 26, the AHR was evaluated using FlexiVent (SCIREQ, Montreal, QC, Canada) following tracheostomy, with PBS or methacholine (10, 20, and 40 mg/mL) used as a stimuli (Appendix A).

### 4.3. BALF and Serum Analysis

To collect BALF from the mice, a tracheostomy was performed 48 h after the last CuO NP or OVA challenge, following a previously described method [22]. Sterilized PBS (700 µL) was instilled into the lung tissue and subsequently withdrawn, and this process was repeated twice. The recovered BALF samples were centrifuged at 300× *g* for 10 min at 4 °C, and the supernatants were collected in fresh EP tubes to assess the levels of IL-4, IL-5, IL-13, IL-1β, IL-6, and TNF-α using commercial enzyme-linked immunosorbent assay (ELISA) kits (BD Biosciences, San Jose, CA, USA). After removing the supernatants, the pellets from the BALF samples were resuspended in 600 µL of PBS for inflammatory cell counting. The total number of cells in the BALF was determined using an automated cell counter (Cell Countess III, Thermo Fisher Scientific, Waltham, MA, USA). For differential cell counting, the resuspended BALF pellets were attached to slide glasses using a cytospin at 200× *g* (Hanil Science, Seoul, Republic of Korea), followed by drying and staining with Diff–Quik solution (ThermoFisher Scientific) according to the manufacturer’s instructions. Differential inflammatory cells were counted using a light microscope at 200× magnification. After collecting BALF samples, whole blood was obtained from the caudal vena cava and centrifuged at 7000× *g* for 10 min to obtain serum. The serum was then used to measure the levels of total and OVA-specific IgE (BioLegend, San Diego, CA, USA).

### 4.4. Histopathology

A histopathological examination was performed on the lung samples collected from the animals as previously described [43]. After collecting the BALF samples, the left lung tissue was fixed using 10% neutral buffered formalin. After 72 h of fixation, the tissues were paraffin-embedded, sectioned at a thickness of 4 µm, and stained with hematoxylin and eosin (Sigma-Aldrich) to assess airway inflammation, and with periodic acid–Schiff solution (IMEB Inc., San Marcos, CA, USA) to evaluate mucus secretion.

### 4.5. IHC and TUNEL Assay

For IHC, the paraffin-embedded tissues were processed as previously described [44]. The primary antibodies used to assess protein expression included anti-TXNIP (1:200 dilution; Novus Biologicals, Littleton, CO, USA) and anti-cleaved caspase-3 (Cell Signaling Technology, Danvers, MA, USA). All the quantitative analyses related to lung tissue were conducted using the image analyzer software Image J, version 1.51. A TUNEL analysis was performed using a kit (Millipore, Billerica, MA, USA) in accordance with the manufacturer’s instructions. Apoptotic cells were randomly counted at 200× magnification in four different fields per sample.

### 4.6. Western Blot Analysis

We performed immunoblotting as described in a previous study [44]. The following primary antibodies and dilutions were used: TXNIP (1:1000; Novus Biologicals); phosphorylated ASK1 (p-ASK1, 1:1000; Cell Signaling Technology); total ASK1 (t-ASK1, 1:1000; Abcam, Cambridge, UK); Bax (1:1000; Cell Signaling Technology); Bcl-2 (1:1000; Cell Signaling Technology); cleaved caspase-3 (1:1000; Cell Signaling Technology); and β-actin (1:1000; Cell Signaling Technology). The secondary antibodies and dilutions used were goat anti-mouse IgG (1:10,000; Thermo Fisher Scientific) and goat anti-rabbit IgG (1:10,000; Thermo Fisher Scientific). The relative densitometric values of the proteins were determined using Chemi-Doc (Bio-Rad Laboratories, Hercules, CA, USA).

### 4.7. Cell Culture and Cell Viability Assay

NCI-H292 cells, a human airway epithelial cell line (American Type Culture Collection, Manassas, VA, USA), were cultivated in an RPMI 1640 medium (WELGENE, Gyungsan, Republic of Korea) supplemented with 10% fetal bovine serum and antibiotics. The cells were incubated in a humidified chamber maintained at 37 °C with 5% CO_2_. Cell viability was measured using the EZ-Cytox cell viability assay kit (Dogenbio, Seoul, Republic of Korea), as previously described [31]. Cells (4 × 10^4^ cells/well) were seeded in a 96-well plate (SPL Life Science, Pocheon, Republic of Korea). After 24 h, a fresh medium or various concentrations of CuO NPs (0.25, 0.5, 1.0, 2.0, and 4.0 μg/mL) were added, and the cells were incubated for 24 h. Subsequently, 10 μL/well of the kit reagent were added to each well, and the plates were incubated for 4 h. The absorbance was measured at 450 nm using an ELISA reader (Bio-Rad Laboratories).

### 4.8. Pro-Inflammatory Cytokine and mRNA Expression Measurement in NCI-H292 Cells

Cells (8 × 10^5^ cells/dish) were seeded in a 60 × 15 mm cell culture dish (SPL Life Science) and cultured for 24 h. Subsequently, the cells were treated with a fresh medium or various concentrations of CuO NPs (0.25, 0.5, 1.0, and 2.0 μg/mL) and incubated for an additional 24 h. The levels of IL-6 and IL-8 in the culture medium were then measured using ELISA kits (R&D Systems, Minneapolis, MN, USA) following the manufacturer’s protocol. After harvesting, RNA extraction was conducted using the HiGene Total RNA Prep Kit (Biofact, Daejeon, Republic of Korea) in accordance with the manufacturer’s instructions, and the extracted total RNA was reverse transcribed into cDNA using a cDNA kit (Qiagen, Hilden, Germany). To measure the mRNA expression of inflammatory cytokines, the qRT-PCR technique previously described [31] was employed. A quantitative analysis was conducted using the real-time PCR detection system (Bio-Rad Laboratories). The qRT-PCR assays were conducted utilizing particular forward and reverse primers, as shown in Appendix A.

### 4.9. Immunofluorescence and Confocal Microscopy

Cells (2 × 10^5^ cells/well) were seeded in a 12-well culture plate (SPL Life Science) and cultured for 24 h, then treated with a fresh medium, CuO NPs (2 μg/mL), or hydrogen peroxide (100 μM) for an additional 12 h. Double immunofluorescence was performed as described in a previous study [22]. The utilized antibodies were as follows: TXNIP (1:200; Novus Biologicals); p-ASK1 (1:200; Cell Signaling Technology); FITC antibody (Rabbit IgG; 1:100; Sigma-Aldrich); and TRITC antibody (Mouse IgG; 1:100; Sigma-Aldrich). Fluorescence images were captured under 630× magnification, using a confocal microscope (ZEISS, Oberkochen, Germany).

### 4.10. Small Interfering RNA Transfection

Cells (8 × 10^5^ cells/well) were seeded for 24 h. Following the manufacturer’s instructions, TXNIP and scrambled siRNA were transfected into NCI-H292 cells using the Lipofectamine™ RNAiMAX reagent (Invitrogen, Waltham, MA, USA). After blocking the expression of TXNIP, the cells were treated with the highest concentration of CuO NPs or with a free medium and were extracted after 6 h.

### 4.11. Statistical Analysis

Data were expressed as the mean ± standard deviation. A one-way ANOVA analysis of variance was used to test for statistical significance among the experimental groups, followed by Tukey’s multiple comparisons test. *p*-values less than 0.05 were considered statistically significant.

## 5. Conclusions

In the present study, CuO NP exposure markedly induces pulmonary toxicity and exacerbates asthma by increasing inflammatory cell infiltration and inflammatory cytokine levels, accompanied by the activation of the TXNIP signaling pathway. These findings suggest that CuO NP exposure may induce pulmonary toxicity and exacerbate asthma through the TXNIP signaling pathway. Thus, this study provides notable insights on the mechanisms involved in pulmonary toxicity and asthma exacerbation induced by CuO NPs, offering novel perspectives on their targeting signal strategies.

## Figures and Tables

**Figure 1 ijms-25-11436-f001:**
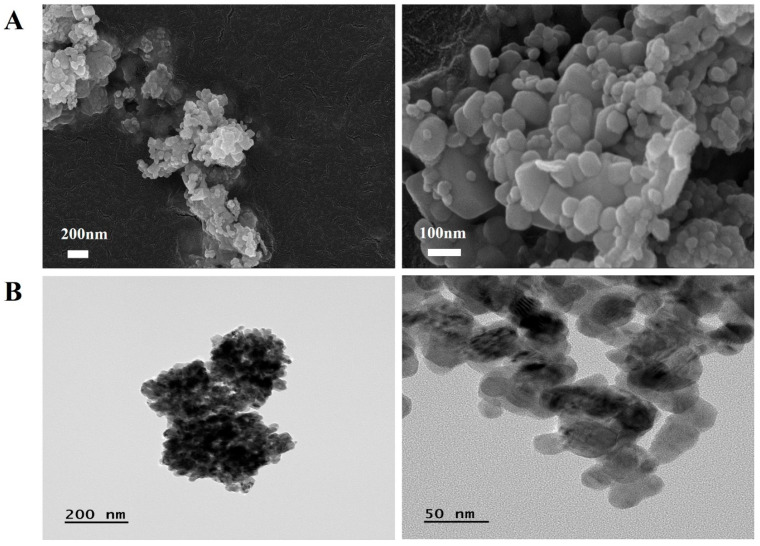
Morphology of CuO NPs. (**A**) The morphology of CuO NPs was measured using scanning electric microscopy (Bar = 200 and 100 nm) and (**B**) transmission electron microscopy (Bar = 200 and 50 nm).

**Figure 2 ijms-25-11436-f002:**
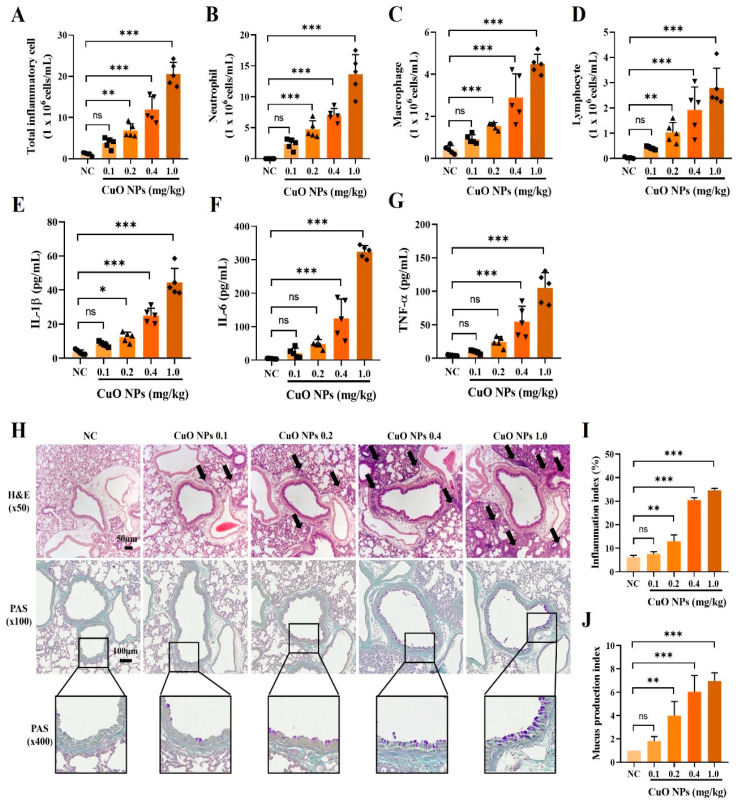
Pathophysiological alterations in mice exposed to CuO NPs. (**A**–**D**) Inflammatory cell counts in BALF. (**E**–**G**) Inflammatory cytokines in BALF. (**H**) Representative figure for lung tissue stained with hematoxylin and eosin (×50, Bar = 50 μm) and staining with periodic acid–Schiff (×100 and ×400, Bar = 100 μm). (**I**,**J**) Quantitative analysis of the inflammatory infiltration and mucus secretion, respectively. The black arrows represent inflammatory cell infiltration. Mice were necropsied 48 h after the last dose of CuO NPs. Data are indicated as mean ± SD (*n* = 5). Significant differences from the NC group are shown by the following symbols: * *p* < 0.05, ** *p* < 0.01, and *** *p* < 0.001; “ns” indicates not significant (*p* > 0.05).

**Figure 3 ijms-25-11436-f003:**
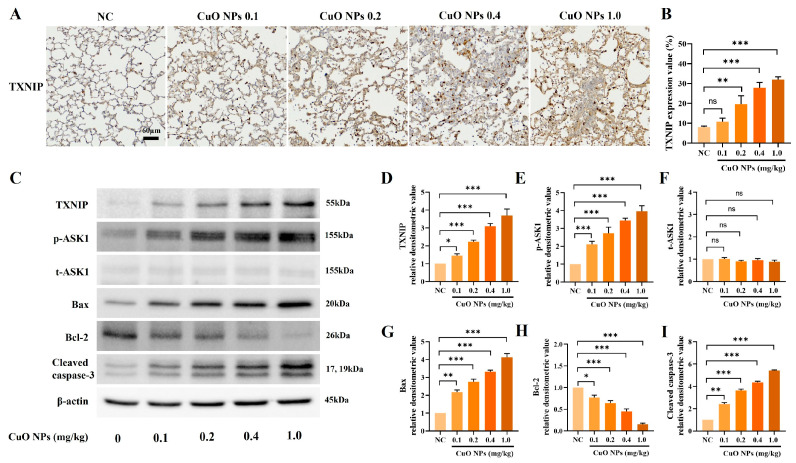
Effects of CuO NP treatment on the protein expression of TXNIP, p-ASK1, total ASK1 (t-ASK1), Bax, Bcl-2, and cleaved caspase-3. (**A**) Representative figure of TXNIP expression on lung tissue via ICH (×200, Bar = 60 nm). (**B**) Expression value of TXNIP. (**C**) Protein expression detected on the Western blot gels. (**D**–**I**) Relative densitometric values of each protein. Mice were necropsied 48 h after the last dose of CuO NPs. Data are indicated as mean ± SD (*n* = 3) Significant differences from the control group are shown by the following symbols: * *p* < 0.05, ** *p* < 0.01, and *** *p* < 0.001; “ns” indicates not significant (*p* > 0.05).

**Figure 4 ijms-25-11436-f004:**
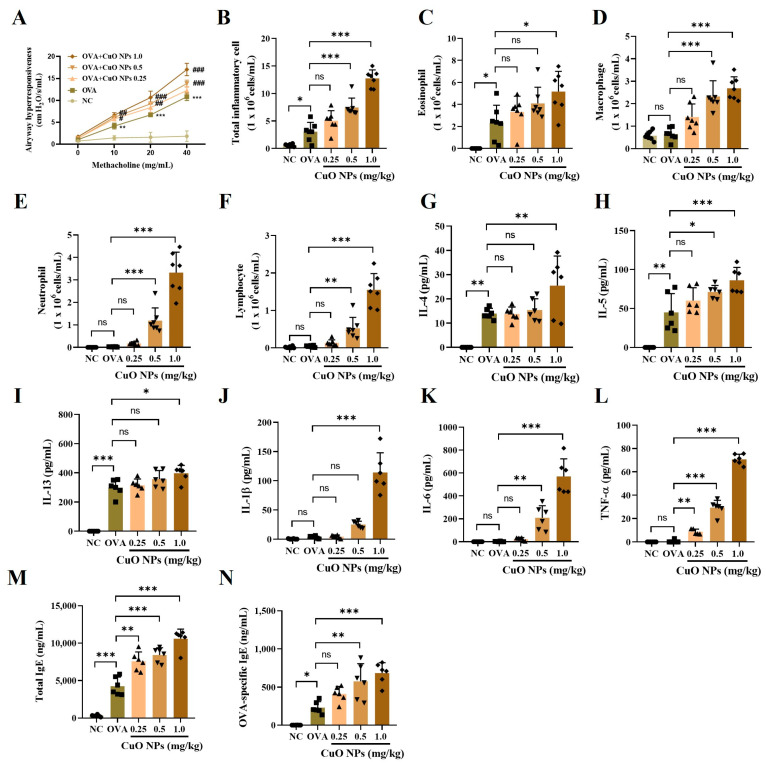
Effects of CuO NP exposure on pathophysiological alterations in asthmatic mice. (**A**) Airway hyperresponsiveness. (**B**–**F**) Inflammatory cell counts in BALF. (**G**–**L**) Inflammatory cytokines in BALF. (**M**,**N**) Total IgE and OVA-specific IgE levels, respectively. Mice were necropsied 48 h after the last OVA challenge. Data are indicated as mean ± SD (*n* = 6). Significant differences are shown by the following symbols: * *p* < 0.05, ** *p* < 0.01, and *** *p* < 0.001. ^#^ *p* < 0.05, ^##^ *p* < 0.01, and ^###^ *p* < 0.001; “ns” indicates not significant (*p* > 0.05).

**Figure 5 ijms-25-11436-f005:**
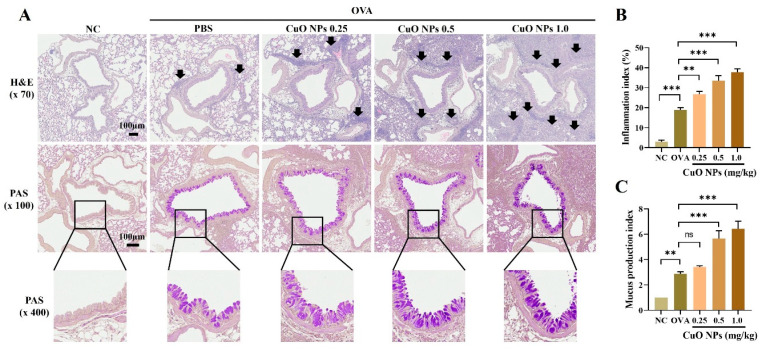
Effects of CuO NP exposure on inflammatory cell infiltration and mucus secretion in the lungs of asthmatic mice. (**A**) Representative figure for lung tissue stained with hematoxylin and eosin (×70, Bar = 100 μm) and periodic acid–Schiff (×100, ×400, Bar = 100 μm). (**B**,**C**) Quantitative analysis of the inflammatory infiltration and mucus secretion, respectively. The black arrows represent inflammatory cell infiltration. Mice were necropsied 48 h after the last OVA challenge. Data are indicated as mean ± SD (*n* = 3). Significant differences are shown by the following symbols: ** *p* < 0.01 and *** *p* < 0.001; “ns” indicates not significant (*p* > 0.05).

**Figure 6 ijms-25-11436-f006:**
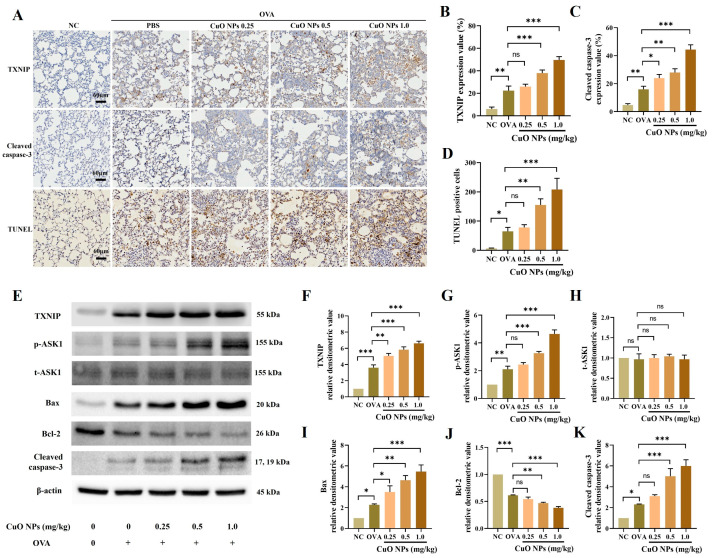
Effects of CuO NP exposure on the TXNIP signaling pathway in asthmatic mice. (**A**) Representative figure for the expression of TXNIP, cleaved caspase-3 (×200, Bar = 60 nm), and the TUNEL assay. (**B**–**D**) Quantitative analyses of protein expression and TUNEL positive cells. (**E**) Protein expression detected on the Western blot gels. (**F**–**K**) Relative densitometric values of each protein. Mice were necropsied 48 h after the last OVA challenge. Data are indicated as mean ± SD (*n* = 3). Significant differences are shown by the following symbols: * *p* < 0.05, ** *p* < 0.01, and *** *p* < 0.001; “ns” indicates not significant (*p* > 0.05).

**Figure 7 ijms-25-11436-f007:**
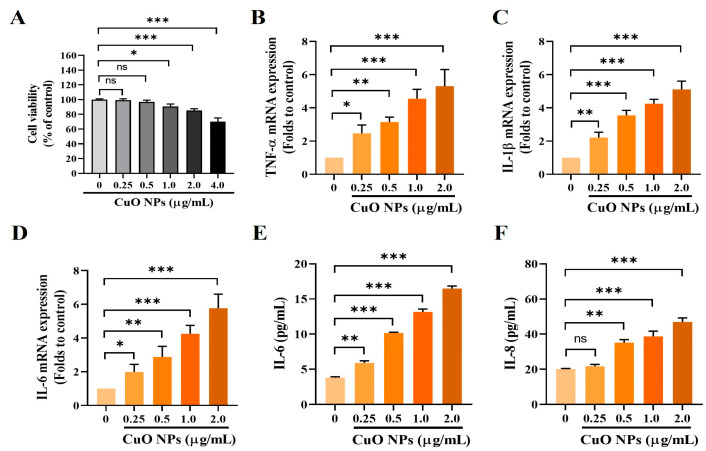
Effects of CuO NP treatment on inflammatory mediators in NCI-H292 cells. (**A**) Cell viability after 24 h incubation following CuO NP treatment. (**B**–**D**) mRNA expression levels of TNF-α, IL-β, and IL-6 measured using real-time PCR, respectively. (**E**,**F**) IL-6 and IL-8 levels determined using ELISA, respectively. The cells were treated with CuO NPs for 24 h. Data are indicated as mean ± SD (*n* = 3). Significant differences from the control group are shown by the following symbols: * *p* < 0.05, ** *p* < 0.01, and *** *p* < 0.001; “ns” indicates not significant (*p* > 0.05). Control, free medium treatment; CuO NP 0.25, 0.5, 1.0, and 2.0, CuO NP treatment (0.25, 0.5, 1.0, and 2.0 μg/mL, respectively).

**Figure 8 ijms-25-11436-f008:**
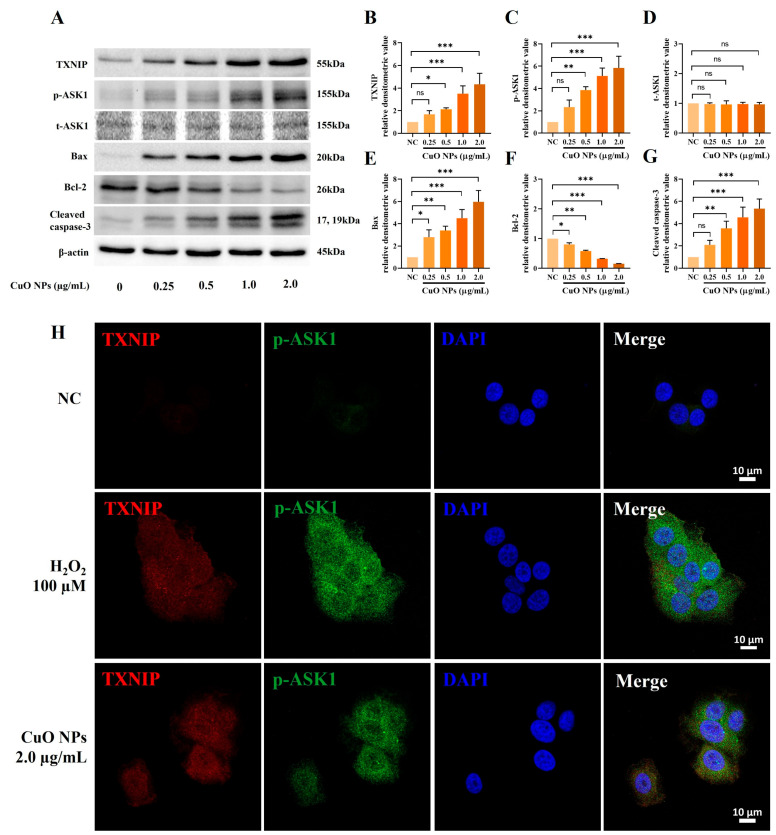
Effects of CuO NP treatment on the TXNIP signaling pathway in NCI-H292 cells. (**A**) Protein expression detected on the Western blot gels. (**B**–**G**) Relative densitometric values of each protein. (**H**) Representative figure for TXNIP and p-ASK1 expression by double-immunofluorescence staining in cells treated with hydrogen peroxide (100 μM), CuO NPs (2.0 μg/mL), and an untreated control (Bar = 10 μm). The cells were treated with a free medium, hydrogen peroxide, or CuO NPs (0.25, 0.5, 1.0, and 2.0 μg/mL) for 12 h. Data are indicated as mean ± SD (*n* = 3). Significant differences from the control group are shown by the following symbols: * *p* < 0.05, ** *p* < 0.01, and *** *p* < 0.001; “ns” indicates not significant (*p* > 0.05).

**Figure 9 ijms-25-11436-f009:**
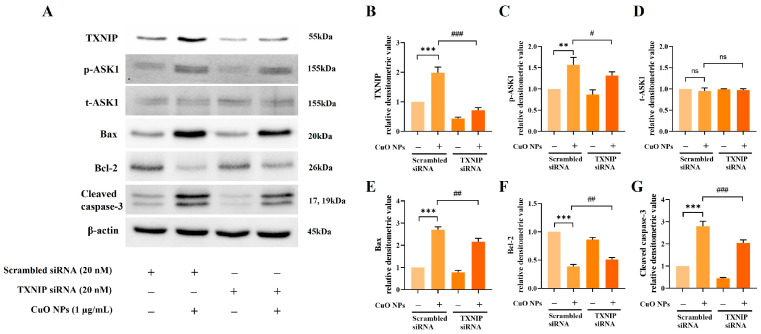
Effects of the downregulation of TXNIP on CuO NP-induced apoptotic signaling pathways in NCI-H292 cells. (**A**) Protein expression detected on the Western blot gels. (**B**–**G**) Relative densitometric values of each protein. The cells were treated with CuO NPs (0.25, 0.5, 1.0, and 2.0 μg/mL) for 6 h, and then harvested. Data are indicated as mean ± SD (*n* = 3). Significant differences from the scrambled siRNA group and scrambled siRNA + CuO NP group are shown by the following symbols: ** *p* < 0.01, *** *p* < 0.001 and ^#^ *p* < 0.05, ^##^ *p* < 0.01, and ^###^ *p* < 0.001, respectively; “ns” indicates not significant (*p* > 0.05).

## Data Availability

Data is contained within the article or Appendix A.

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
