# Peer review of "Copper Oxide Nanoparticles Induce Pulmonary Inflammation and Exacerbate Asthma via the TXNIP Signaling Pathway"

_ijms, 2024, doi:10.3390/ijms252111436_

Round 1
Reviewer 1 Report
Comments and Suggestions for Authors
Kim et al investigated the lung toxicity of copper oxide nanoparticles (CuO NPs) and their effect on OVA-induced asthma. Previous studies have already shown that CuO NPs are toxic to lung epithelial cells. The novelty of the present work is that the TXNIP pathway plays a decisive role in the toxicity of CuO NPs. However, the Western blot (Wb) results presented are inconclusive. Based on the original Wb images, doubts arise as to whether the same amount of samples were applied to the gel in each test. Concerns could have been allayed if a load control had been added to every gel.
The manuscript also contains a number of inaccuracies. The sample number in Figure 2 is incorrectly stated (n=5). The description of the controls is completely missing in some cases (Figure 5). The title of Figure 6 is inaccurate and inconsistent with the results presented.
Comments on the Quality of English LanguageThe English of the manuscript is adequate, although the wording of some sentences is imprecise (for example: "Protein expression on the gels.", "Effects of CuO NPs treatment on inflammatory mediators of NCI-H292 Cells.")
Author Response
Original comments, black; Revised works, Red
#Reviewer 1
Comments 1: Kim et al investigated the lung toxicity of copper oxide nanoparticles (CuO NPs) and their effect on OVA-induced asthma. Previous studies have already shown that CuO NPs are toxic to lung epithelial cells. The novelty of the present work is that the TXNIP pathway plays a decisive role in the toxicity of CuO NPs. However, the Western blot (Wb) results presented are inconclusive. Based on the original Wb images, doubts arise as to whether the same amount of samples were applied to the gel in each test. Concerns could have been allayed if a load control had been added to every gel.
Response 1: Thank you for your insightful comment and for highlighting the importance of consistent loading controls in validating Western blot results. We understand your concern regarding the equal loading of samples across different gels. To address this, we have included the original Western blot images in the PDF file, where β-actin was used as the loading control for each gel. For each set, we quantified 30 µg of sample protein in a total volume of 80 µl and performed Western blotting by loading 10 µl per lane for a total of six different factors (TXNIP, p-ASK1, t-ASK1, Bax, Bcl-2, Cleaved caspase-3) and β-actin. This ensured that β-actin was consistently loaded in every gel, serving as a reliable loading control. Additionally, we rearranged the placement of the figures within the PDF to make it easier to observe the loading controls and protein expressions at a glance. As shown in the original images, β-actin expression levels were nearly uniform across all lanes, confirming that equal amounts of protein were applied. We hope this clarifies your concern and confirms the reliability of our data.
Comments 2: The manuscript also contains a number of inaccuracies. The sample number in Figure 2 is incorrectly stated (n=5). The description of the controls is completely missing in some cases (Figure 5). The title of Figure 6 is inaccurate and inconsistent with the results presented.
Response 2: Thank you for your careful review and for pointing out the inaccuracies and inconsistencies in the manuscript. We have corrected the sample number in Figure 2 (Line 116) and updated the figure 5 legend (now Figure 8 in the revised manuscript) and result to ensure that the controls are clearly explained. Additionally, we have revised the results section for Figure 6 (now Figure 9 in the revised manuscript) and updated the title in the Figure 6 (now Figure 9 in the revised manuscript) legend to accurately reflect the findings. We appreciate your attention to these details.
“Additionally, double-immunofluorescence staining revealed a marked accumulation of TXNIP and p-ASK1 in CuO NPs-treated cells (2.0 µg/mL) compared to untreated cells. This accumulation was comparable to that observed in hydrogen peroxide-treated cells (100 µM), which serves as a known inducer of apoptosis (Figure 8H).” (Page 9: Lines 217–221)
“Representative figure for TXNIP and p-ASK1 expression by double-immunofluorescence staining in cells treated with hydrogen peroxide (100 μM), CuO NPs (2.0 μg/mL), and untreated control (Bar = 10 μm).” (Page 10: Line 225–227)
" To investigate whether TXNIP influences the regulation of apoptotic signaling pathways in CuO NPs-treated NCI-H292 cells, we conducted a functional analysis using TXNIP gene knockdown via siRNA. TXNIP siRNA treatment significantly reduced the elevated expression levels of p-ASK1, Bax, and cleaved caspase-3 induced by CuO NPs exposure compared to the scrambled siRNA treatment, while the expression level of Bcl-2 was correspondingly increased (Figure 9)." (Page 10: Lines 232–237)
"Effects of downregulation of TXNIP on CuO NPs-induced apoptotic signaling pathways in NCI-H292 cells." (Page 11: Lines 239–240)
Comments 3: The English of the manuscript is adequate, although the wording of some sentences is imprecise (for example: "Protein expression on the gels.", "Effects of CuO NPs treatment on inflammatory mediators of NCI-H292 Cells.")
Response 3: Thank you for your valuable feedback. We have carefully revised the wording of the sentences you highlighted to improve clarity and accuracy.
“Protein expression detected on the Western blot gels.” (Page 5: Lines 128–129, Page 8: 192–193, Page 10: 223–224, Page 11: 240)
“Effects of CuO NPs treatment on inflammatory mediators in NCI-H292 Cells.” (Page 9: Line 207)
Reviewer 2 Report
Comments and Suggestions for Authors
The result section must be improved. The information provided regarding how the experiments were performed is scarce. For instance, information regarding the CuO NP incubation times or treatment time in in vitro and in vivo experiments is lacking. This information should be incorporated into the figure legends.
In addition, the time point of the in vivo experiment performed to evaluate the acute effect of CuO NPs in not asthma mice was not specified in the materials and methods sections either.
Section 2.4. lines 123-124: 2.0 ug/mL is the maximum concentration of CuO NPs tested? According to what is shown in Figure 4A, the maximum concentration is 4.0 ug/mL.
The authors should have calculated the IC50 of CuO NPs, even if the cytokine production on the cell line used was below the concentration of 4.0 ug/mL, which is the CuO NP concentration in which these NPs start to be cytotoxic.
The result shown in Figure H is poorly described, a comparison between the positive control (cells treated with H2O2 100 uM) and the CuO incubated cells is missing.
I suggest reordering the results and grouping the first in vivo set of results with the last ones (that include the group of mice sensitized with OVA).
Comments on the Quality of English Language
The quality of English should be improved, I found typos and some grammar mistakes.
Author Response
Original comments, black; Revised works, Red
# Reviewer 2
Comments 1: The result section must be improved. The information provided regarding how the experiments were performed is scarce. For instance, information regarding the CuO NP incubation times or treatment time in in vitro and in vivo experiments is lacking. This information should be incorporated into the figure legends.
Response 1: Thank you for your valuable feedback. Following your suggestion, we have provided more detailed information about the experimental procedure in the figure legends to offer greater clarity of the experimental conditions. We have also included a more detailed description of the cells, and CuO NPs treatments in the materials and methods section. The specific details added are as follows:
"Mice were necropsied 48 hours after the last dose of CuO NPs. " (Page 4: Lines 115–116, Page 5: Lines 129–130)
"Mice were necropsied 48 hours after the last OVA challenge." (Page 6: Lines 153–154, Page 7: Lines 170, Page 8: Line 193–194)
"The cells were treated with CuO NPs (0.25, 0.5, 1.0 and 2.0 μg/mL) for 24 h." (Page 9: Lines 210–211)
"The cells were treated with free medium or hydrogen peroxide or CuO NPs (0.25, 0.5, 1.0 and 2.0 μg/mL) for 12 h." (Page 10: Lines 227–228)
"The cells were treated with CuO NPs (0.25, 0.5, 1.0 and 2.0 μg/mL) for 6 h, and then harvested." (Page 11: Lines 241–242)
"Cells (8 × 10⁵ cells/well) were seeded in a 60 × 15 mm cell culture dish (SPL Life Science) and cultured for 24 h. Subsequently, the cells were treated with fresh medium or various concentrations of CuO NPs (0.25, 0.5, 1.0, and 2.0 μg/mL) and incubated for an additional 24 h. " (Page 14: Lines 405–408)
"Cells (2 × 10⁵ cells/well) were seeded in a 12-well culture plate (SPL Life Science) and cultured for 24 h, then treated with fresh medium, CuO NPs (2 μg/mL), or hydrogen per-oxide (100 μM) for an additional 24 h." (Page 14: Lines 419–421)
"Cells (8 × 105 cells/well) were seeded for 24 h." (Page 14: Line 428)
Comments 2: In addition, the time point of the in vivo experiment performed to evaluate the acute effect of CuO NPs in not asthma mice was not specified in the materials and methods sections either.
Response 2: Thank you for your valuable comment. We have revised the Materials and Methods section to specify the time point of the in vivo experiment performed to evaluate the acute effects of CuO NPs in non-asthmatic mice. Additionally, this information is also provided in figure legends and Supplementary Figure 1 for further clarification.
Figure S1. Illustration showing the experimental design. (A) CuO NPs-induced pulmonary toxicity experimental schedule. Mice were necropsied 48 hours after the last dose of CuO NPs. (B) CuO NPs-induced asthma exacerbation experimental schedule. Mice were necropsied 48 hours after the last OVA challenge.
“Mice were necropsied 48 hours after the last CuO NPs dose for subsequent analyses (Figure S1A).” (Page 13: Lines 336–337)
“To collect BALF from the mice, tracheostomy was performed 48 h after the last CuO NPs or OVA challenge, following a previously described method.” (Page 13: Lines 349–350)
Comments 3: Section 2.4. lines 123-124: 2.0 ug/mL is the maximum concentration of CuO NPs tested? According to what is shown in Figure 4A, the maximum concentration is 4.0 ug/mL.
Response 3: As shown in Figure 4A (now Figure 7A in the revised manuscript), we chose 2 µg/mL as the highest concentration for our cell experiments based on the results of the cell viability assay. Although concentrations up to 4 µg/mL were initially tested, we observed that concentrations above 2 µg/mL resulted in excessive cellular damage, which skew the data and not accurately reflect the physiological effects of CuO NPs. Since our in vitro experiments aimed to identify changes in inflammatory mediators and mechanisms induced by CuO NPs, we selected 2 µg/mL as the highest concentration. At this concentration, cell viability remained at approximately 80%, which we considered optimal for evaluating the biological effects of CuO NPs with minimal cytotoxic interference. Additionally, this concentration is consistent with those used in previous studies, ensuring alignment with established research [1,2]. To further clarify this rationale, we have revised the manuscript as follows:
References:
- Ko, J.W.; Park, J.W.; Shin, N.R.; Kim, J.H.; Cho, Y.K.; Shin, D.H.; Kim, J.C.; Lee, I.C.; Oh, S.R.; Ahn, K.S.; et al. Copper oxide nanoparticle induces inflammatory response and mucus production via mapk signaling in human bronchial epithelial cells. Environ Toxicol Pharmacol 2016, 43, 21–26.
- Ko, J.W.; Shin, N.R.; Park, J.W.; Park, S.H.; Lee, I.C.; Kim, J.S.; Kim, J.C.; Ahn, K.S.; Shin, I.S. Copper oxide nanoparticles induce collagen deposition via tgf-β1/smad3 signaling in human airway epithelial cells. Nanotoxicology 2018, 12 (3), 239–250.
"Based on the cell viability assay, the maximum concentration of CuO NPs for in vitro experiments was set at 2.0 µg/mL, as higher concentrations resulted in significant cytotoxicity (Figure 7A)." (Page 8: Lines 198–200)
Comments 4: The authors should have calculated the IC50 of CuO NPs, even if the cytokine production on the cell line used was below the concentration of 4.0 ug/mL, which is the CuO NP concentration in which these NPs start to be cytotoxic.
Response 4: Thank you for your valuable feedback. Based on your comments, we have calculated the IC₅₀ to gain further insight into the cytotoxicity profile of CuO NPs, and the data are presented below. However, given the scope of our study and experimental design, our primary objective was to investigate the effects of CuO NPs on cytokine production without excessive cytotoxic interference, in order to better understand the underlying mechanisms. Therefore, we focused on a concentration of 2 µg/mL, which maintained approximately 80% cell viability (IC₂₀ value). We believe that the cell viability assay provides a reliable and relevant assessment of the cytotoxicity of CuO NPs within the experimental parameters we established. This approach aligns with the IC₂₀ used to assess physiological effects in previous studies [1–3], and we are confident that the current data sufficiently capture the effects of CuO NPs on both cell viability and cytokine production. We sincerely hope that this explanation clarifies our approach and highlights the valuable insights this study provides within the investigated concentration range.
References:
- Banjerdpongchai, R.; Khawon, P.; Pompimon, W. Phytochemicals from goniothalamus griffithii induce human cancer cell apoptosis. Asian Pac. J. Cancer Prev. 2016, 17 (7), 3281–3287.
- Bastos, V.; Duarte, I.F.; Santos, C.; Oliveira, H. A study of the effects of citrate-coated silver nanoparticles on raw 264.7 cells using a toolbox of cytotoxic endpoints. Journal of Nanoparticle Research 2017, 19 (5), 163.
- Kim, Y.-J.; Yang, S.I.; Ryu, J.-C. Cytotoxicity and genotoxicity of nano-silver in mammalian cell lines. Molecular & Cellular Toxicology 2010, 6 (2), 119–125.
Comments 5: The result shown in Figure H is poorly described, a comparison between the positive control (cells treated with H2O2 100 uM) and the CuO incubated cells is missing.
Response 5: Thank you for your valuable comment. As per your suggestion, we have revised the results in Figure 8H to include a comparison with the positive control. Specifically, we added the following statement:
"Additionally, double-immunofluorescence staining revealed a marked accumulation of TXNIP and p-ASK1 in CuO NPs-treated cells (2.0 µg/mL) compared to untreated cells. This accumulation was comparable to that observed in hydrogen peroxide-treated cells (100 µM), which serves as a known inducer of apoptosis (Figure 8H).” (Page 9: Lines 217–221)
Comments 6: I suggest reordering the results and grouping the first in vivo set of results with the last ones (that include the group of mice sensitized with OVA).
Response 6: Thank you for your valuable suggestion regarding the reordering of the results. We understand the importance of presenting the findings in a logical and coherent manner. Based on your feedback, we have revised the sequence of the results to group the in vivo findings together, followed by the in vitro results. The updated order is as follows: in vivo general CuO NPs toxicity, in vivo CuO NPs toxicity in the asthmatic mouse model, and finally, the in vitro cytotoxicity experiments. Specifically, Section 2.7 has been moved to 2.4, Section 2.8 to 2.5, and Section 2.9 to 2.6. Meanwhile, the original Sections 2.4, 2.5, and 2.6 have been adjusted to 2.7, 2.8, and 2.9, respectively. We believe that this revised structure maintains the logical flow of the study while also aligning with your suggestion, thereby providing a clearer understanding of the overall effects of CuO NPs in both non-asthmatic and asthmatic conditions. We hope this reorganization addresses your concerns and enhances the clarity of our manuscript.
Comments 7: The quality of English should be improved, I found typos and some grammar mistakes.
Response 7: We have carefully reviewed the entire manuscript and made corrections to typos and grammatical mistakes to improve the clarity and accuracy of the language. We hope that the revised version meets the required standards, and we sincerely appreciate your attention to detail.
Reviewer 3 Report
Comments and Suggestions for Authors
In the manuscript entitled "Copper Oxide Nanoparticles Exacerbate Allergic Responses in an Ovalbumin-induced Asthma Model via Elevation of the TXNIP Signaling Pathway", the authors investigate the effects of CuO NPs on allergic response in vivo and in vitro. The TXNIP signaling was found to contribute to the toxic effects of CuO NPs. This study is intresting, however, the manuscript should be improved before further processing. The following concerns should be addressed in the revised version of the manuscript.
- In vivo model and cell experiment were used in the study, improve or remove "in an Ovalbumin-induced Asthma Model" will be better.
- Fig 4A, statistical mark is missing.
- The cytokins should be detected using ELISA.
- Apoptosis should be detected in the in vitro study.
- Fig 5, the displayed data suggests that the cells have undergone apoptosis, but the morphology of the nucleus appears normal.
- For the Caspase 3 expression in western boltting data, how was it analyzed?
Author Response
In the manuscript entitled "Copper Oxide Nanoparticles Exacerbate Allergic Responses in an Ovalbumin-induced Asthma Model via Elevation of the TXNIP Signaling Pathway", the authors investigate the effects of CuO NPs on allergic response in vivo and in vitro. The TXNIP signaling was found to contribute to the toxic effects of CuO NPs. This study is intresting, however, the manuscript should be improved before further processing. The following concerns should be addressed in the revised version of the manuscript.
Comments 1: In vivo model and cell experiment were used in the study, improve or remove "in an Ovalbumin-induced Asthma Model" will be better.
Response 1: Thank you for your valuable suggestion regarding the manuscript title. Based on your feedback, we have revised the original title, 'Copper Oxide Nanoparticles Exacerbate Allergic Responses in an OVA-induced Asthma Model via Elevation of the TXNIP Signaling Pathway,' to 'Copper Oxide Nanoparticles Induce Pulmonary Inflammation and Exacerbate Asthma via the TXNIP Signaling Pathway' to more accurately reflect the focus and findings of our study. We believe that this revised title more effectively captures the core aspects of our research.
“Copper Oxide Nanoparticles Induce Pulmonary Inflammation and Exacerbate Asthma via the TXNIP signaling pathway” (Page 1: Lines 2–3)
Comments 2: Fig 4A, statistical mark is missing.
Response 2: Thank you for pointing out the missing statistical marks in Fig 4A (now Figure 7A in the revised manuscript). We have added the appropriate statistical comparisons to provide a clearer understanding of the results.
Comments 3: The cytokins should be detected using ELISA.
Response 3: Thank you for your insightful comment regarding the detection of cytokines using ELISA. In our vitro investigation, we successfully measured IL-6 and IL-8 protein levels using ELISA, as shown in our results (Figure 7). However, despite multiple attempts using kits from different manufacturers, TNF-α and IL-1β protein levels were consistently undetectable because they remained below the detection limits under our experimental conditions. To still provide meaningful insights into the effects of CuO NPs, we chose to present the mRNA expression levels of TNF-α and IL-1β as an alternative measure. These mRNA data serve to indicate the transcriptional changes induced by CuO NPs exposure. We acknowledge that protein level measurements would provide more direct evidence and will consider optimizing conditions for future experiments. We hope this explanation clarifies our approach, and we appreciate your understanding.
Comments 4: Apoptosis should be detected in the in vitro study.
Response 4: Thank you for your valuable suggestions regarding the detection of apoptosis in our in vitro study. While apoptosis can be detected through direct methods such as annexin V/PI staining or electron microscopy, as well as through Western blotting to detect apoptotic proteins [1], we were unable to employ these direct detection methods due to practical limitations in our laboratory environment. Instead, we focused on monitoring apoptosis through Western blotting, which is a widely accepted and indirect method for identifying apoptotic changes. Moreover, given that the primary aim of our cell experiments was to study inflammatory markers and TXNIP-related mechanisms, we selected this approach to identify proteins indicative of apoptotic activity. In Figure 8A-G, the Western blot analysis demonstrates the expression of cleaved caspase-3, a key marker of apoptosis, which increased in a dose-dependent manner with CuO NP treatment. Additionally, changes in the expression of pro-apoptotic (Bax) and anti-apoptotic (Bcl-2) proteins further support the induction of apoptosis by CuO NPs. Furthermore, in Figure 8H, double-immunofluorescence staining revealed a significant accumulation of TXNIP and phosphorylated ASK1 (p-ASK1) in CuO NP-treated cells (2.0 µg/mL) compared to untreated controls. This pattern was comparable to the positive control, hydrogen peroxide (100 µM), a well-known apoptosis inducer. Since TXNIP can activate ASK1 to promote apoptosis, the observed increase in p-ASK1 supports the conclusion that CuO NPs induced apoptosis in our in vitro model. We hope this explanation clarifies our approach and addresses your concern, given both the focus of our experiments and the constraints we faced.
References:
- Banfalvi, G. Methods to detect apoptotic cell death. Apoptosis 2017, 22 (2), 306–323.
Comments 5: Fig 5, the displayed data suggests that the cells have undergone apoptosis, but the morphology of the nucleus appears normal.
Response 5: Thank you for your insightful comment. In this study, we used 100 μM hydrogen peroxide (H₂O₂) as a positive control. Although H₂O₂ is known to effectively induce apoptosis, its effectiveness can vary significantly depending on the concentration used. As reported in previous studies [1–3], 100 μM of H₂O₂ does not always cause significant cytological changes. The primary goal of our in vitro experiments was to investigate changes in inflammatory markers and the activation of TXNIP-related signaling pathways, rather than to focus on observing cytological changes. Therefore, we selected a CuO NPs concentration that minimized excessive cell death, allowing us to observe signaling pathway changes more clearly. Our in vitro results suggest that the process of cell death had been initiated but was not yet complete at the concentration used. We hope this explanation clarifies our approach and rationale behind our choice of concentrations, and that it adequately addresses your concerns.
References:
- Kim, D.K.; Cho, E.S.; Um, H.D. Caspase-dependent and -independent events in apoptosis induced by hydrogen peroxide. Exp. Cell Res. 2000, 257 (1), 82–88.
- Nakajima, Y.; Aoshiba, K.; Yasui, S.; Nagai, A. H2O2 induces apoptosis in bovine tracheal epithelial cells in vitro. Life Sci. 1999, 64 (26), 2489–2496.
- Xiang, J.; Wan, C.; Guo, R.; Guo, D. Is hydrogen peroxide a suitable apoptosis inducer for all cell types? Biomed Res Int 2016, 2016, 7343965.
Comments 6: For the Caspase 3 expression in western boltting data, how was it analyzed?
Response 6: First, we sincerely apologize for the oversight in notation. The Western blotting experiments in our study were conducted to detect cleaved caspase-3, not caspase-3, and the notation error was an unfortunate mistake in the manuscript. We have corrected all instances throughout the manuscript to accurately indicate "cleaved caspase-3" instead of "caspase-3" to reflect the proper analysis. For the experiments, cleaved caspase-3 was detected using a specific antibody from Cell Signaling Technology (#9664), and the relative density values were quantified using the Chemi-Doc imaging system (Bio-Rad Laboratories). We hope that this correction clarifies any confusion caused by the initial oversight.
Round 2
Reviewer 1 Report
Comments and Suggestions for Authors
I still maintain that there is something wrong with the way Western blots are performed or evaluated. Let's look at just one example! Fig 3. p-ASK1 Western blot images: How specific is the antibody that reacts with so many proteins? Why is the pattern of the protein so different in 1st and 2-3rd gel? If the amount of loaded proteins was the same, how is it possible that the amount of all the proteins recognized by the antibody, not just the 155 kD protein, increases from left to right in the gels? The same doubts arise for many other gels.
Author Response
Comments 1: I still maintain that there is something wrong with the way Western blots are performed or evaluated. Let's look at just one example! Fig 3. p-ASK1 Western blot images: How specific is the antibody that reacts with so many proteins? Why is the pattern of the protein so different in 1st and 2-3rd gel? If the amount of loaded proteins was the same, how is it possible that the amount of all the proteins recognized by the antibody, not just the 155 kD protein, increases from left to right in the gels? The same doubts arise for many other gels.
Response 1: Thank you for your detailed feedback and for highlighting these potential concerns. In response to your comments, the co-authors have conducted a thorough review and discussion of the experimental procedures.
We understand your concerns regarding antibody specificity. We used commercially available antibodies that have been validated in previous studies (e.g., Cell Signaling Technology, Novus, Abcam), and these are specific to their defined target sizes. However, factors such as tissue composition and antibody binding kinetics can introduce variability, particularly in complex biological samples like those from in vivo models, and non-specific binding to sample lysates can occasionally occur. We acknowledge this issue and will take additional steps to minimize non-specific binding in future studies by refining blotting conditions, including adjusting the washing times and durations, carefully selecting appropriate blocking reagents, and optimizing antibody concentrations.
Your concerns about the inconsistencies in gel patterns and protein loading are also valid. While we followed the same antibody and sample preparation protocols, and ensured that the same amount of total protein was loaded into each well, we believe that slight procedural variations, such as experimenter error, differences in protein transfer efficiency, and loading precision, may have contributed to these discrepancies. Although these factors are small, they can sometimes lead to inconsistencies, particularly in signal intensity. Moving forward, we will introduce more thorough controls and further standardize our sample preparation protocols to reduce these variations.
Despite the aforementioned limitations, we were able to demonstrate a clear increase in the expression of the target protein, which supports the main findings of this study. However, we recognize that further optimization is needed to strengthen these results. In future studies, we will work to improve our experimental protocols to ensure more consistent and accurate detection, and we are confident that these improvements will enhance the reliability of our data in future experiments.
We hope this response addresses the issues you raised, and we sincerely thank you again for your valuable feedback, which will help us refine our approach moving forward.
Reviewer 2 Report
Comments and Suggestions for Authors
The authors replied correctly to all reviewer questions and added the suggested modifications to the article. Thus, I recommend this article be published
Author Response
Thank you for your valuable review and effort.
Reviewer 3 Report
Comments and Suggestions for Authors
The labeling of caspase3 is still confusing. What are the two bands and is the antibody specifically detecting cleaved caspase3? The caspase3 labeled in Figure 8A, and the other figures are cleaned caspase3.
Author Response
Comments 1: The labeling of caspase3 is still confusing. What are the two bands and is the antibody specifically detecting cleaved caspase3? The caspase3 labeled in Figure 8A, and the other figures are cleaned caspase3.
Response 1: We sincerely apologize for the confusion caused by the labeling of caspase-3 in Figure 8A. While we have revised the manuscript to accurately reflect cleaved caspase-3 throughout the text, we regret that the original image was used in Figure 8A, which contributed to this misunderstanding. We have now corrected this issue.
As previously mentioned, the antibody used for our experiments was ‘Cleaved Caspase-3 (Asp175) (5A1E) Rabbit mAb #9664’ from Cell Signaling Technology. According to the data sheet provided by the manufacturer, this antibody specifically detects cleaved caspase-3 at 17,19kDa (https://www.cellsignal.com/products/9664/datasheet?images=1&protocol=0&size=A4). Additionally, this antibody has been referenced in various studies for its specificity in detecting cleaved caspase-3 [1–3]. We hope this clarification resolves any confusion, and we appreciate your understanding.
References:
- Zhou, B.; Wang, L.; Yang, S.; Liang, Y.; Zhang, Y.; Pan, X.; Li, J. Rosmarinic acid treatment protects against lethal h1n1 virus-mediated inflammation and lung injury by promoting activation of the h-pgds-pgd(2)-ho-1 signal axis. Chin. Med. 2023, 18 (1), 139.
- Wu, X.J.; Zhang, Z.; Wong, J.P.; Rivera-Soto, R.; White, M.C.; Rai, A.A.; Damania, B. Kaposi's sarcoma-associated herpesvirus viral protein kinase augments cell survival. Cell Death Dis. 2023, 14 (10), 688.
- Lim, J.O.; Kim, W.I.; Pak, S.W.; Lee, S.J.; Moon, C.; Shin, I.S.; Kim, S.H.; Kim, J.C. Pycnogenol-assisted alleviation of titanium dioxide nanoparticle-induced lung inflammation via thioredoxin-interacting protein downregulation. Antioxidants (Basel) 2024, 13 (8), 972.
Round 3
Reviewer 1 Report
Comments and Suggestions for Authors
The authors' answer did not dispel my doubts.
Author Response
Thank you for your detailed feedback and for highlighting these potential concerns. Based on your comments, we have conducted a thorough review and discussion of the experimental procedures and added a description of cleaved caspases.